# Immature Testicular Tissue Engineered from Weaned Mice to Adults for Prepubertal Fertility Preservation—An In Vivo Translational Study

**DOI:** 10.3390/ijms23042042

**Published:** 2022-02-12

**Authors:** How Tseng, Yung-Liang Liu, Buo-Jia Lu, Chi-Huang Chen

**Affiliations:** 1Department of Biochemistry and Molecular Cell Biology, School of Medicine, College of Medicine, Taipei Medical University, Taipei 110, Taiwan; tsenghow@tmu.edu.tw; 2Graduate Institute of Medical Sciences, College of Medicine, Taipei Medical University, Taipei 110, Taiwan; 3International Ph.D. Program for Cell Therapy and Regeneration Medicine, College of Medicine, Taipei Medical University, Taipei 110, Taiwan; 4Department of Obstetrics and Gynecology, Chung Shan Medical University Hospital, Taichung 40203, Taiwan; h121976@yahoo.com.tw; 5Division of Reproductive Medicine, Department of Obstetrics and Gynecology, Taipei Medical University Hospital, Taipei 110, Taiwan; beckcha050121@gmail.com; 6Department of Obstetrics and Gynecology, School of Medicine, College of Medicine, Taipei Medical University, Taipei 110, Taiwan

**Keywords:** fertility preservation, immature testis tissue (ITT), poly-l-lactic acid (PLLA), scaffold, transgenic mouse, bioluminescent imaging (BLI), tissue engineering, spermatogenesis

## Abstract

Male pediatric survivors of cancers and bone marrow transplantation often require adjuvant chemoradiation therapy that may be gonadotoxic. The optimal methods to preserve fertility in these prepubertal males are still under investigation. This manuscript presents an in vivo experiment which involved transplantation of immature testicular tissues (ITT) from transgenic donor, to wild-type recipient mice. Donors and recipients were age-mismatched (from 20-week-old donors to 3-week-old recipients, and vice versa) and the transplantation sites involved the abdomen, skin of the head, back muscle, and scrotum. The application of poly-l-lactic acid (PLLA) scaffold was also evaluated in age-matched donors and recipients (both 3-weeks-old). To quantitively evaluate the process of spermatogenesis after ITT transplantation and scaffold application, bioluminescence imaging (BLI) was employed. Our result showed that ITT from 3-week-old mice had the best potential for spermatogenesis, and the optimal transplantation site was in the scrotum. Spermatogenesis was observed in recipient mice up to 51 days after transplantation, and up to the 85th day if scaffold was used. The peak of spermatogenesis occurred between the 42nd and 55th days in the scaffold group. This animal model may serve as a framework for further studies in prepubertal male fertility preservation.

## 1. Introduction

Many of the current fertility preservation methods for sexually immature male cancer patients and non-malignant hematologic disorders before hematopoietic stem cell transplant (HSCT), are at risk of gonadotoxic chemoradiotherapy and are still experimental [1,2,3]. Regarding the molecular mechanism of male germ cell damage, the deleterious effects of alkylating agents, radiotherapy, air pollution and heavy metals in biological systems are mainly mediated through the generation of DNA oxidative damage and cause variable degrees of germ cell death in a time and dose–dependent manner, and subsequently result in subfertility or permanent infertility [4,5,6].The most ethically acceptable and increasingly adopted method that can be offered to prepubertal boys at present, is the use of immature testicular tissues (ITT). ITT contain spermatogonial stem cells (SSCs) that can self-renew, and some studies have achieved successful resumption of spermatogenesis in animals after autografting, or in vitro maturation of fresh, cryopreserved, or vitrified ITT [7,8,9,10,11,12,13].

A central strategy for the regeneration of immature tissues or the survival of tissue grafts niches is to reverse the suboptimal environment caused by the wound, fibrosis, or surgical and medical interventions. The different components of the stem cell niche are complex and dynamic, requiring a combination of direct and indirect interactions, such as secreted factors, inflammation and scarring, extracellular matrix (ECM), physical parameters, and environmental signals, between cells [14].

ECM is a critical part of the niche of almost all tissues. With current advances in tissue engineering, ECM may serve as an alternative method to conventional transplantation methods through the creation of new, polymeric biomaterials as an artificial scaffold. Poly-l-lactic acid (PLLA), an aliphatic polyester, is one of the most promising polymers and scaffold materials used in tissue engineering research and has the potential to mimic the natural ECM in the future. It is biodegradable, biocompatible, minimally inflammatory, mechanical and thermoplastic properties make PLLA a suitable resorbable implant material, and it is already in use in many medical applications [15,16,17].

To date, however, there is a paucity of studies using PLLA-based scaffold to apply to ITT transplantation. Although it is well known that ECM both anchors stem cells and directs their fate, the incomplete knowledge of the prepubertal SSC niche and the conditions required for differentiation, hinders the development of suitable transplantation protocols [18,19]. In addition, there is a lack of convenient and time-efficient methods to elucidate human relevant processes by tracking in vivo tissue and cell events involved in spermatogenesis, or for rapid evaluations of alternative assays for their usefulness or effectiveness.

In our previous study, we used FVB/N-Tg (*PolII-luc*) Ltc transgenic mice that served as a bridge to establish protocols or surrogate studies of pertinent processes, for human germline and stem cell studies [20]. The method, which was based on bioluminescence imaging (BLI) of the transgenic mouse model to track the survival of ovarian tissues, could measure the success of different immunosuppressive regimens, cryopreservation protocols, and transplantation methods, commonly used in infertility research [21,22,23]. The mouse isograft showed no obvious signs of rejection, which was similar to other organs [24]. Taken together, ITT has rarely been applied to the technique of in vivo tissue engineering for prepubertal fertility preservation.

The current multidisciplinary study aimed to determine the effectiveness of applying a fibrous, microscale scaffold to increase graft survival using ITTs with transgenic mice as a surrogate model for iso-transplantation, which mimics auto-transplantation based on bioengineering and translational medicine. This attempt showed the feasibility of using the advantages of BLI for in vivo and longitudinal ITT fate mapping of spermatogenesis from sexually immature mice to adults, by gonadal tissue engineering (Figure 1).

Isolated testis from the FVB/N-Tg (*PolII-luc*) Ltc transgenic donor mice. Attached mashed testis tissue on electrospun PLLA mats, instantly isograft into wild type recipient mice. Luciferin was injected before observing mice with an IVIS system.

## 2. Results

### 2.1. Different Ages of Donor Testicular Tissues Transplanted into Various Recipient Sites

The testicular tissues of donors (FVB/N-Tg) from 20- and 3-week-old mice were grafted into the scrotum, head, abdomen, or back skin of 3-week (wt) and 20-week-old recipients (wt), respectively, and development was tracked using BLI for 51 days. Based on the QY of the BLI analysis shown in situ, the scrotum was determined to be the optimal site for testicular tissue development; similarly, younger donors were better than older donors (Figure 2).

### 2.2. Long-Term In Vivo Tracking of ITT Spermatogenesis in Age-Matched Donors and Recipients with/without Scaffold Bioengineering

Immature transgenic donor testicular tissue was transplanted to age-matched recipient scrotum with or without a PLLA scaffold. Testicular tissues from three-week-old donors were transplanted into the post-orchiectomy scrotum of age-matched recipients. Comparison with or without a PLLA scaffold showed that the grafted testicular tissues with scaffold could increase the degree of sperm proliferation compared to those without the scaffold, in particular, on the 5th, 7th, and 42nd days after transplantation (* *p* < 0.05) (Figure 3A); in a two-week period, from the 42nd to the 55th day, the grafted ITT reached a peak and a plateau in spermatogenesis. From the 55th day to the 85th day, spermatogenesis gradually declined but maintained a higher level than the fresh control (Figure 3B).

### 2.3. Histological Assays of Spermatogenesis of the ITT Graft

After removal of long-term engraftment of the mature testicular tissues, the hematoxylin and eosin (HE) stain showed significantly thicker seminiferous tubules in the scaffold group (Figure 4B), compared with the control group (Figure 4A), bar = 100 µm. The magnified box of the scaffold group indicates the existent of PLLA scaffold (the arrows between scaffolds, Figure 4B). The arrowhead shows the significant spermatogenesis with mature sperm marked in yellow circle in the seminiferous tubules in scaffold group (Figure 4D) and the control group (4C). The Immunohistochemistry (IHC) staining showed FVB/N-Tg (*PolII-luc*) Ltc transgenic donor testicular tissue in both groups (Figure 4E,F) and SYCP3-positivity, required for normal meiosis during spermatogenesis, in both the groups (Figure 4G,H), bar = 100 µm.

## 3. Discussion

This work was based on a multidisciplinary study design, combining mouse ITT iso-transplantation to mimic auto-transplantation, long-term BLI tracking in vivo, and PLLA-based tissue engineering. Emulating the local ECM and the ability to monitor cellular and tissue responses to bioengineered microenvironments in vivo, are paramount for achieving greater success rates in the regeneration of tissues to yield spermatogenesis for fertility preservation in immature mice. A translational model in small primates is a prerequisite to application in human pediatric patients, which could take years, or decades, before transplanting back, preserved ITT, to regain fertility.

In a recent study, ITT transplantation has been successfully achieved in a rhesus monkey, followed by live birth of offspring via assisted reproductive technology, which offers hope for fertility preservation in prepubertal boys who are about to undergo cancer treatments [25]. Our study, while similar, differs in two aspects: it offers a longitudinal, in vivo observation of ITT engineering, and grafts were all transplanted into the scrotum rather than beneath the skin. Via in vivo fate mapping, it was observed that ITT engineered with the scaffold, facilitated progressive spermatogenesis, which reached a peak and a plateau during a critical two-week period. By mimicking the physiological conditions of the in vivo microenvironment, our study seeks to provide a better understanding of PLLA scaffold as a valid ECM substitute.

The use of BLI in vivo allows researchers to observe biological processes within the organism as a whole and may accelerate the transition away from cell culture systems. Furthermore, this tool may facilitate tissue engineering as a translational model before human application [26]. This real-time, in vivo experiment using optical imaging, provides more advantages than in vitro studies [22,26]. BLI has shown its versatility in adapting the use of reporter genes in various clinical applications. Undoubtedly, BLI played a vital role in determining the optimal transplantation site(s) and donor tissue progression in vivo in our studies.

The data on graft survival when transplanted to different parts of the recipient mice demonstrated the importance of the niche. The quantification of emitted light was followed for up to 51 days (Figure 2) and showed the scrotum to be the optimal grafting site to receive testis tissue from both young and older transgenic donor mice. The donor tissues survived in all the graft sites. Our in vivo imaging profile is similar to a previous study, which obtained complete spermatogenesis in marmosets only when autologous grafts of testicular tissue were transplanted into the scrotum, but not under the skin [27]. Xenotransplantation of human ITT from human fetuses or prepubertal boys has not yet produced complete spermatogenesis [28,29,30]. Neither did the addition of antioxidants, follicle-stimulating hormone, or testosterone improve the outcome of xenografts [30,31,32,33]. Our initial data and the studies by Luetjens et al., [27] and others previously mentioned, highlight the fact that appropriate host conditions are critical for transplant success. The regenerative capabilities might differ across species and the phylogenetic distances between species may also be a limiting factor in the success of xenotransplantation [19,27]. We suggest that the best efficiency may be achieved by autologous transplantation of frozen-thawed ITT in situ in humans.

We also investigated age differences among donor and recipient mice in transplantation success rates as others have also addressed the potential effects of age-dependent differences in humans, pigs, and bovines [19,30,34,35,36]. In our BLI, brightness curve data (Figure 2) indicated that younger tissues appeared to be better suited than older ones. Younger tissue with more spermatogonial stem cells emitted higher QY that persisted throughout the experiment in almost all areas of the transplantation sites to indicate the optimal site is the scrotum in situ (Figure 2). Similarly, a study by Caires et al., on the xenotransplantation of piglet testicular tissue of different maturities, to castrated nude mice showed that tissue from younger donors showed a higher capacity for germ and Sertoli cells to achieve complete differentiation after grafting [35]. Recovery of human SSC (fetal versus adult human tissue) or ITT (in two age groups) after long-term cryopreservation or xenografting, also depended on the age of the human testicular tissues or single-cell suspensions [30,36]. Age-related differences can play an essential part in the ability of grafted tissue to survive and differentiate in the host environment, possibly due to the greater regenerative potential of younger testicular tissues that harbor more stem cells. Similar observations were seen with xenografted bovine tissue with different stages of maturity expressing differential gene expression and, thus, subsequently produced different proteins [19,34]. The results of our studies have significant implications regarding the appropriate recipient environment and age-dependent differences, thus, highlighting the complex interaction between niche and age-related graft survival and regenerative capacities, but this requires further investigation.

Our subsequent study aimed to investigate if the addition of scaffold to immature, donor testicular tissues after transplantation, increases survival rates compared with tissue-only grafts. Randomly arranged, PLLA micro-scaffold was selected as the substrate for our study because the morphology and architecture of the electrospun structure was used to mimic the natural ECM and has been shown to enhance cellular infiltration [37,38,39]. The subject of interest is the time period immediately after tissue or organ transplantation, when grafts are often exposed to significant time periods of tissue hypoxia and ischemia. Lee et al., in a comparison of PLLA of different topographies, observed that randomly oriented PLLA fibers exhibited a higher cellular infiltration and promoted blood vessel growth possibly as a result of its relatively large surface-to-volume ratio [39]. Similarly, our randomly arranged PLLA scaffold used in this experiment also showed the ability to support cell and tissue growth in the initial critical period of initial ischemia until the tissues can produce their own physiological matrix environment and blood vessels.

BLI imaging was used concurrently to provide a convenient, real-time monitoring of graft survival that shows much better immature graft survival, especially the crucial spermatogenesis peak between the 42nd and 55th days in the scaffold group. BLI declines, but is still better than the control group up to the 85th day (Figure 3B). It is interesting to note that the critical peak spermatogenesis occurred between the 42nd and 55th days. All the mice were weaned before the age of four weeks. Most mice (males and females) reach sexual maturity at 4–7 weeks and, therefore, are not typically mated until they reach 6–8 weeks of age. The disruption in transportation of the partial seminiferous tubules of ITTs is accompanied by spermatogenesis.

Male germ cell apoptosis, in general, includes developmental and induced cell death because germ cell apoptosis is involved in every step of testicular development in nature. It is plausible to speculate that the spermatogenesis within the disconnected seminiferous tubules without transportation to epididymis may lead to sperm apoptosis, increased reactive oxygen species and sperm death. The sperm output changed dynamically as it eventually reached a limit within the confined space of seminiferous tubules, which may cause tissues to adjust to cellular competition to adapt to increased architectural complexity over time [40]. Of note, biodegradation and biocompatibility of PLLA can last as long as 180 days [41], thereby allowing us to utilize it for long-term tracking of tissue-engineered spermatogenesis from prepubertal to adult mice until a steady level in vivo for 85 days.

In comparison, the scaffold groups more closely resemble normal testicular tissue ECM and appear to have a higher density of cells and sperm in addition to higher counts of sperm. Histological studies mirror those of the BLI images, suggesting that scaffolds promoted enhanced cellular growth and graft survival. Both the histology and IHC support the fact that significantly better spermatogenesis occurred in ITT with a PLLA scaffold (Figure 4).

Improvement in scaffold materials combined with the outstanding shaping and molding properties of PLLA means that the polymers can be formed in various sizes, shapes, organizations, and topographical structures with different fiber diameters and pore sizes [16,42]. The membrane can also be coated with synthetic or natural biopolymers for a variety of applications [43], including the future potential of nanoscale or 3-D printing of ITT matrix with a close resemblance to ITT grafts.

This study does have some limitations. As mentioned in our previous studies, although our transgenic model can be tracked and quantified for viable grafts, it cannot pinpoint the stages of cellular differentiation. Tracking each stage of tissue differentiation may be made possible with future designs of tissue-specific promoters to help fine-tune grafting protocols [43].

Taken together, BLI technology was combined with ITT engineering and end-point analysis to monitor ITT grafts in vivo. This surrogate mouse model employing cryopreserved ITT, supports the potential for prepubertal male fertility preservation in humans. ITT engineering enhanced the regenerative capacity of testicular tissues and critical spermatogenesis in the optimal site of the scrotum, younger ITT, and scaffold application. The in vivo fate mapping of critical peak spermatogenesis of mouse ITT grafts, from the prepubertal to adult ages, led to a two-week spermatogenesis peak after sexual maturation, followed by a decline, possibly due to the limited ITT capacity, which provides new insights and merits for human application.

## 4. Materials and Methods

### 4.1. In Vivo Study

The ITT donors were a transgenic (Tg) mouse line (FVB/N-Tg (*PolII-luc*) Ltc) with the H_2_^q^ haploid genotype created by the Level Transgenic Center of Level Biotechnology Inc. (New Taipei City, Taiwan) (Figure 5). The details regarding donor and recipient mice were described in our previous studies [20,21,23].

The housing and breeding conditions for the mice in the animal house were under 22–24 °C and followed a 12/12 h light/dark regimen without restriction of food and water supplies. The forementioned procedures were approved by the Animal Experimental Committee at the Taipei Medical University (Taipei, Taiwan) and adhered to the Guide for the Care and Use of Laboratory Animals (National Institute of Health).

This study was a two-part experiment based on donor FVB/N-Tg (*PolII-luc*) Ltc transgenic mice to recipient FVB/NJNarl wild-type (wt) mice. The first part aimed to determine the optimal transplantation sites for ITT, as well as the spermatogenesis potential of testicular tissue grafts from donor mice of different ages. The second part involved transplanting ITT from 3-week-old donors into the post-orchiectomy scrotum of aged-matched recipients with or without a PLLA scaffold. In both parts of the study, FVB/N-Tg (*PolII-luc*) Ltc transgenic mice and inbred FVB/NJNarl wild-type mice were used as testicular tissue donors and recipients, respectively. All surgical and grafting procedures were performed under general anesthesia induced by intraperitoneal injection of ketamine (50 mg/kg) and xylazine (15 mg/kg). The recipient mice underwent unilateral orchiectomy directly before each experiment.

### 4.2. Scaffold Preparation and Loading of Immature Seminiferous Tubules on the Scaffold 

PLLA membrane was fabricated from PLLA (MW = 140 kDa, glass transition temperature, *T*_g_ = 60–62 °C) purchased from Biotech One Inc. (New Taipei City, Taiwan) by electrospinning. Dichloromethane (CH_2_Cl_2_) was purchased from Acros Organics (Morris Plains, NJ, USA). The preparation of the electrospun membranes to micrometer diameters was adapted from our previous research [16]. Briefly, PLLA was dissolved in CH_2_Cl_2_ at a concentration of 20% (v/v). The electrospinning conditions were as follows: a delivery capillary with an inner diameter of 1 mm with a positive voltage of 15 kV applied at the tip was attached to a syringe pump that fed the fibers at a constant rate of 10 mL/h. A DC motor (composed of a stainless-steel disk, 12.5 cm in diameter) served as a collector and was located 15 cm from the capillary tip. The decellularized testicular albuginea fiber diameter is around 100 nm, and the coarse scaffold diameter is 2.64 ± 0.26 μm. The fabricate electrospun scaffolds mainly focused on fibrous structures analogous to native tissues on testicular tissue reconstruction. 

The scaffold fabricating process is similar to non-woven fabrics, and the fibers may be tightly intertwined or linked. Whether the fibers are connected depends on the solvent vaporizing speed of the electrospinning process. The longer the nozzle, the thinner the fibers are, and the faster the solvent vaporizing speed, the less the connection and the more interweaving. Fibrous linkages could increase scaffold strength, but they also prolong degradation time and most of its characteristics connected a few fibers in an interwoven structure. 

The extraction of testicular tissue was made through surgical methods. The segments were thereby divided into tiny segments and applied to the scaffold. The seminiferous tubules loaded on the single surface of the scaffold, and tissue were freely grafted into the empty scrotum, not fixed. The 3-week-old seminiferous tubules are 0.0135 ± 0.002 mm^2^ in area, 0.461 ± 0.05 mm in perimeter, with a size of 0.129 ± 0.014 mm in diameter. The scaffold is a membrane filled with tiny-sized pores measured at 0.5 cm in diameter, which is adhered to segmented testicular pieces that are approximately 12.2 ± 1.65 mg in weight in contrast to the fresh control of the 14.04 ± 1.62 mg respectively. The loaded seminiferous tubules were not counted by number, exclusively record tissue weight.

### 4.3. Scanning Electron Microscopy (SEM)

The images of the mouse testicular albuginea were taken with a S2400 Tabletop scanning electron microscope (SEM) (Hitachi, Japan) at 5000× magnification (Figure 6A). To validate the diameters of the PLLA scaffold and its morphology, the polymers were studied under the SEM (Figure 6B). The images were taken with a TM3030 Tabletop SEM (Hitachi, Japan) at 1800× magnification at an accelerating voltage of 15 kV. The fiber diameters were measured using commercial imaging software. The pore size distribution of the electrospun membranes was measured with a capillary flow porometer (GFP-1100AX, Porous Materials Inc., USA). Each size of the round PLLA scaffold (0.5 cm in diameter) was used for tissue loading.

### 4.4. In Vivo Bioluminescence Imaging

Mouse testicular tissue grafts were longitudinally tracked in real-time using BLI with the IVIS-200 system, and the luminescence was quantified using the IVIS software. The mice were shaved before each imaging. 150 mg/kg Luciferin (L_8220, Biosynth Carbosynth, Staad, Switzerland) was injected intra-peritoneally into mice 5 min before imaging, and recipients were subsequently placed in a light-tight camera box under general anesthesia with continuous isoflurane (2%). Mice were imaged from the ventral side with a field of view of 20 cm for 3 min at high-resolution settings. A cooled charge-coupled device (CCD) camera performed the BLI with an additional overlay image of black-and-white pictures taken with the aid of a light in the imaging chamber.

The luminescence was quantified by summing the pixel intensities inside the region of interest (ROI; 1.5 cm × 1.5 cm) and absolute light intensity calibrated using an 8-inch integrating sphere (OL series 425 Variable Low-Light-Level Calibration Standard, Optronic Laboratories, Inc., Orlando, FL, USA) as described in [44]. The light emitted by the catalytic reaction of luciferin by luciferase within the ROI was measured by the photon counts on the digitized image, captured by the CCD camera on the integrating sphere. The photon counts were converted to physical units of radiance in photons/s per cm^2^ per steradian [45]. The IVIS software quantified and archived the signals from the images. Digitized BLI images and QY were initially recorded on the same post-transplantation day.

### 4.5. Part I Study: BLI Tracking of the Optimal Graft Sites

In vivo spermatogenesis potential of testicular tissue grafts from donor mice of different age was measured. For this study, donor testicular tissues were sliced into small, thin sections and grafted into four separate areas of recipient. Small pieces of murine testicular tissues were used to increase vascularization and survival. Donor ITT at 3- and 20-weeks-old, and mature testicular tissue were transplanted to 20- and 3-week-old recipients, respectively. Grafted sites included the head, abdomen, back muscle, and scrotum. Graft development was followed with BLI until the day 51 (Figure 2).

### 4.6. Part II Study: BLI Tracking of the ITT Grafts Engineered from PLLA Scaffold

Based on the results gathered in the part I study, the approach to the main study used transplants into the scrotum (of the orchiectomy) of segmented, donor testicular tissues from 3-week-old mice (*n* = 5) to age-matched recipients (*n* = 5) with or without a PLLA scaffold. The tissue weight was 0.014 mg ± 0.002 mg for the control group and 0.012 mg ± 0.002 mg for the experimental group (Figure 3A,B).

Donor tissues were loaded with the PLLA scaffold before grafting into the empty scrotum of recipient mice. Immature testicular development was again followed with BLI up to 85 days after transplantation, using the LIVINGIMAGE v2.50 software (IVIS Software, Caliper Life.

### 4.7. Histology and Immunohistochemistry (IHC) Staining

Grafts were transplanted with or without scaffold. During this part of the study, testicular grafts from recipient mice were removed for histological studies. The testicular tissues were fixed in Bouin’s solution (HT10132, Sigma-Aldrich, Saint Louis, USA) overnight, embedded in paraffin wax, sectioned at 5 μm intervals, then dewaxed and stained with H and E stain. Randomized histological sections were studied from each recipient. Grafted testicular tissue with or without (tissue-only) a PLLA scaffold were used to examine the morphology of the seminiferous tubules and to detect the presence of sperm and differences in sperm density.

IHC staining was performed following previously described protocols [46,47,48]. 5 μm-thick, Bouin-fixed paraffin-embedded tissues were used for IHC staining. The sections were deparaffinized using xylene and were rehydrated in alcohol washes at decreasing concentrations (100%, 100%, 80%, 60%, 50%), rinsed in distilled water, and then exposed to a target retrieval solution (K805, Dako, Santa Clara, CA, USA). The antigen-unmasking solution and slides were heated in an oven at 63 °C for 60 min. The slides were cooled for at least 1 h at room temperature, rinsed with phosphate buffer saline, and any nonspecific sites were subsequently blocked with a blocking solution (S2023, Dako, Santa Clara, CA, USA) for 5 min. The sections were then incubated with a goat polyclonal anti-firefly luciferase antibody (AbCam Inc., Cambridge, UK) and a rabbit polyclonal anti-SYCP3 antibody (Novus Biologicals, Colorado, USA) in an antibody diluent (S3022, Dako, Santa Clara, CA, USA) overnight at 4 °C, followed by a horseradish-peroxidase-conjugated rabbit anti-goat immunoglobulin G (IgG) antibody (Sigma-Aldrich, Saint Louis, MO, USA), and a goat anti-rabbit IgG antibody (Novus Biologicals, Littleton, CO, USA). After washing, the reaction products were assessed with a color solution consisting of a DAB detection system (K5007, Dako, Santa Clara, CA, USA), and the slides were counterstained with H and E. The expression of Firefly Luciferase and SYCP3 was evaluated under a slide scanner, ScanScope (APERIO, Vista, CA, USA).

### 4.8. Statistical Analysis

Two-tailed *t* tests were used for statistical analyses with *p* < 0.05 considered statistically significant.

## 5. Conclusions

This translational bioengineering study may echo the messages from a cutting-edge review article, entitled “Fertility preservation for prepubertal boys: lessons learned from the past and update on remaining challenges towards clinical translation” [49]. Since there has so far been no live births from various fertility preservation methods in prepubertal boys, partly due to ethical concerns as well as the lack of long-term longitudinal follow-ups, our study may pave the way for future applications.

## Figures and Tables

**Figure 1 ijms-23-02042-f001:**
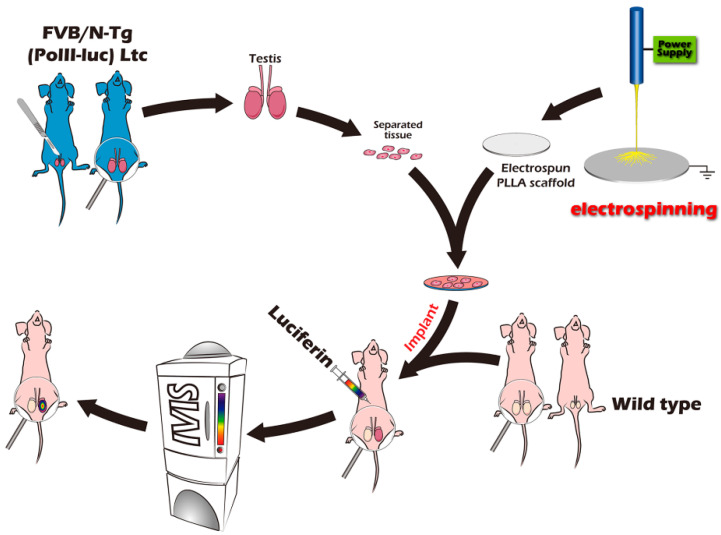
Schematic overview of the study design.

**Figure 2 ijms-23-02042-f002:**
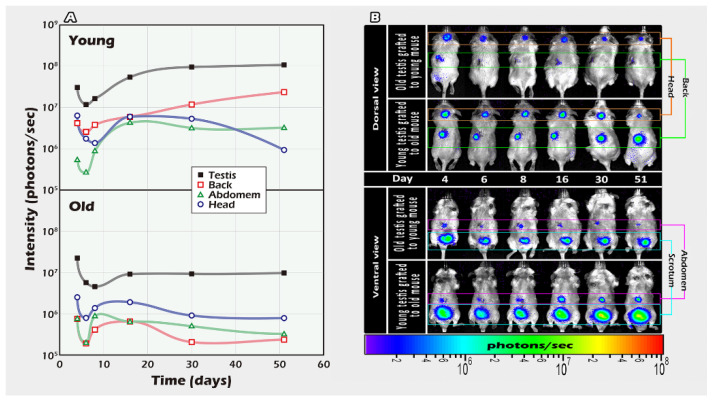
BLI intensities and images of testicular tissue grafts in vivo. The development of the testis tissue of donors (FVB/N-Tg) from 20- or 3-week-old grafted into the scrotum, head, abdomen cavity, or back muscles of 3- and 20-week-old recipients (wt) was tracked using BLI for 51 days. BLI analysis showed that the scrotum is the optimal site for testicular tissue development, younger donors are better than older donors. (**A**) BLI intensity of different graft sites. (**B**) In vivo tracking of the BLI image of ITT engraftment in individuals.

**Figure 3 ijms-23-02042-f003:**
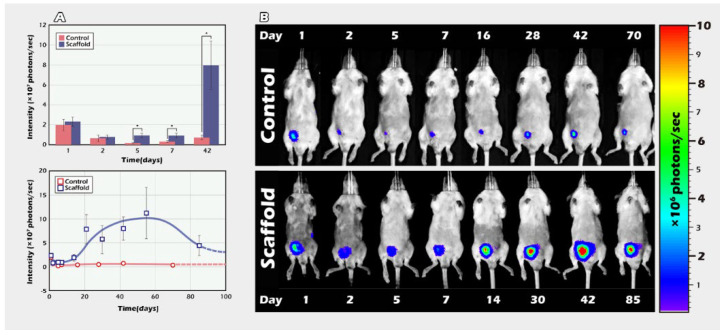
BLI image of 3-week-old, age-matched graft recipients, with or without scaffold until adults in vivo. (**A**) Recipients (3-weeks-old), of transplanted age matched donor testicular tissue, to the scrotum of unilateral orchiectomy with or without a PLLA scaffold. The grafted testis tissue with scaffold could increase the degree of cell regeneration when compared with tissue without the scaffold, in particular, in the short term (5 and 7 days) after transplantation, and also in the long term (42 d) (* *p* < 0.05). Based on the BLI quantity, the signal intensity of immature testicular graft combined with spermatogenesis, increases to a crucial peak between the 42nd and 55th days, then decreases after the 55th day to the end of the 85th tracking day. The quality and quantity of BLI showed significantly effective, peak spermatogenesis, with or without scaffold. (**B**) In vivo tracking of the BLI image showing the typical pattern of the representative in situ ITT engraftment in an individual.

**Figure 4 ijms-23-02042-f004:**
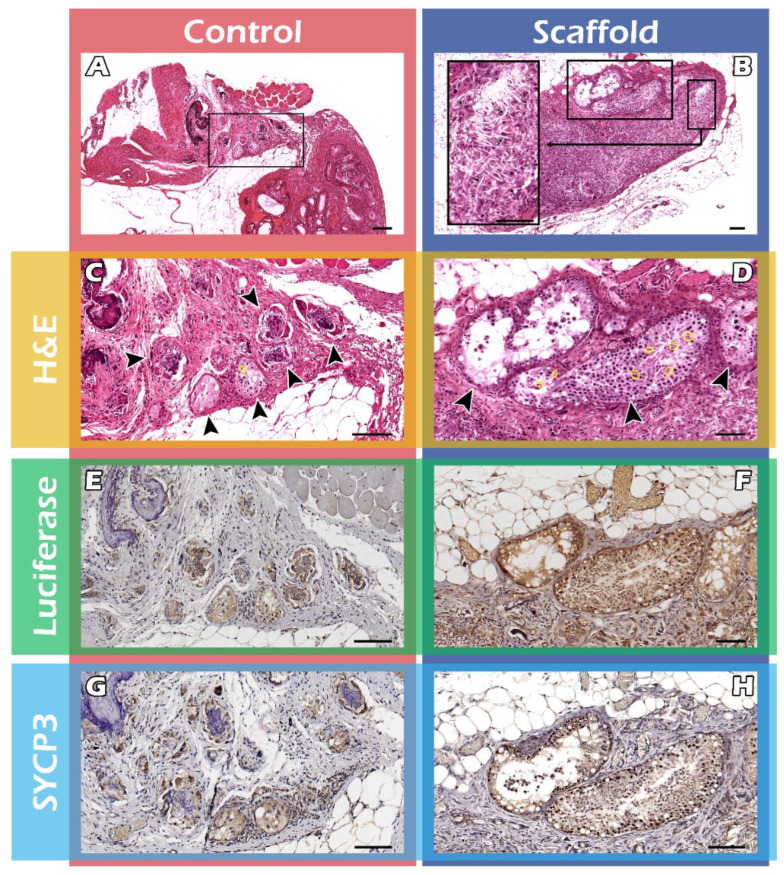
H and E, and IHC stain of testicular grafts. After removal of the long-term engraftment of the mature testicular tissue, the H and E stain shows significantly effective spermatogenesis in the scaffold group (**B**), compared with the control group (**A**) in the seminiferous tubules (arrowhead), bar = 100 μm. The arrows in the magnified box of the scaffold group indicates the existent of PLLA scaffold (**B**). The magnified H and E stain showed spermatogenesis and mature sperm marked in yellow (**C**,**D**) the IHC stain shows the testicular tissue of the FVB/N-Tg (*PolII-luc*) Ltc transgenic donor in both groups (**E**,**F**). Positive SYCP3 stain required for meiosis during spermatogenesis in both groups but significantly better spermatogenesis in the scaffold group (**H**) than control (**G**), bar = 100 μm.

**Figure 5 ijms-23-02042-f005:**
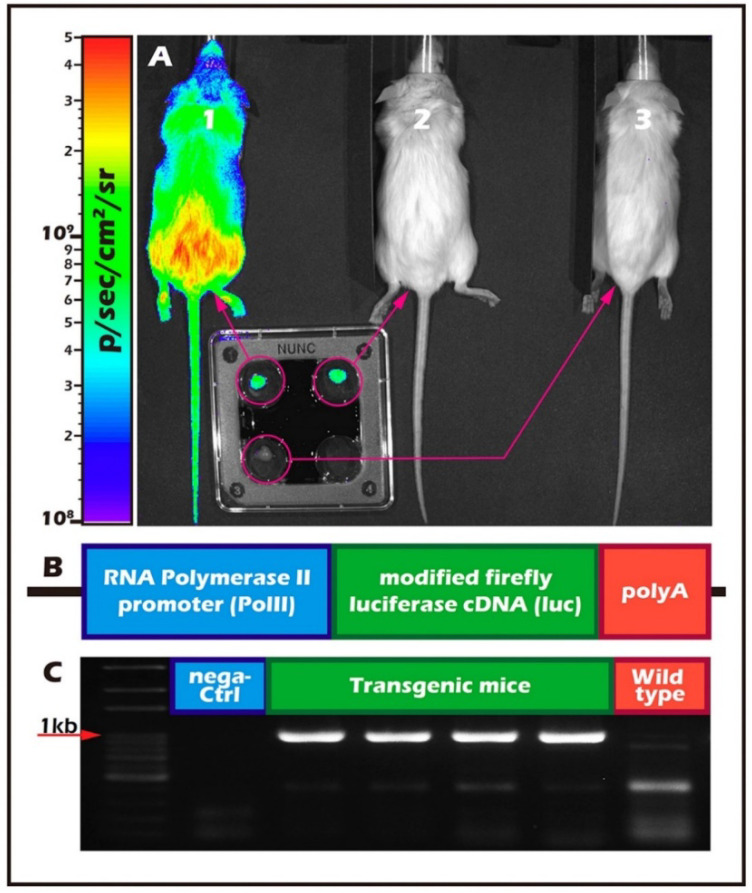
Background of FVB/N-Tg (*PolII-luc*) Ltc transgenic donor mice. (**A**) FVB/N-Tg (*PolII-luc*) Ltc transgenic donor mice (mouse 1) treated with luciferin; the light emitted by the transgene reaction was detected with the IVIS imaging system. The luciferase expression was detectable in all organs of the transgenic mouse, including the isolation of intact testis independently (mouse 2). No light emission was detected from a wild-type donor’s intact testis isolated from the recipient mice (mouse 3). (**B**) Schematic design of the transgenic construct including the mouse RNA polymerase II (*PolII*) promoter and a modified firefly luciferase cDNA (Promega pGL-2). (**C**) Genotyping the transgene by PCR (Polymerase chain reaction).

**Figure 6 ijms-23-02042-f006:**
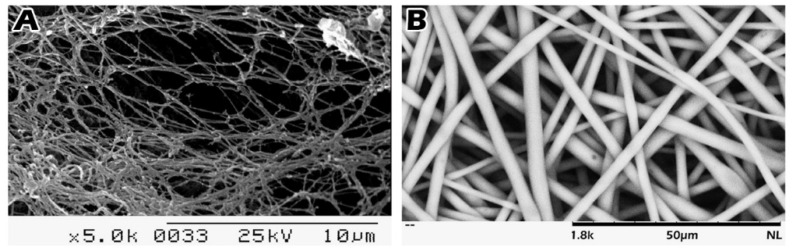
SEM images of matrix of testicular albuginea (**A**) and scaffold (**B**). (**A**) The images of the mouse testicular albuginea taken with a S2400 Tabletop scanning electron microscope (SEM) (Hitachi, Japan) at 5000× magnification. (**B**) To validate the diameters of the PLLA scaffold and its morphology, SEM images were taken with a TM3030 Tabletop SEM (Hitachi, Japan) at 1800× magnification.

## Data Availability

The data that supports the findings of this study are available from the corresponding author upon reasonable request.

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
