# Peer review of "Immature Testicular Tissue Engineered from Weaned Mice to Adults for Prepubertal Fertility Preservation—An In Vivo Translational Study"

_ijms, 2022, doi:10.3390/ijms23042042_

Round 1

Reviewer 1 Report

The manuscript titled “Immature testicular tissue engineered from weaned mice to adults for prepubertal fertility preservation- An in vivo translational study”   it's an in vivo translational study  aimed to determine the effectiveness of applying a fibrous, microscale scaffold to increase graft survival using ITTs with transgenic mice as a surrogate model for iso-transplantation to mimic auto-transplantation based on bioengineering translational medicine. Specifically, the authors use immature testicular tissue engineered from weaned to adult mice for prepubertal fertility preservation.

I thing that this work was carefully conducted but there are some aspect to clarify

Therefore I suggest a major revision

  • Abstract is over 200 words. needs to be shortened and made clearer
  • Better explain why male survivors of pediatric cancers and bone marrow transplantation are at risk for gonadotoxic chemoradiation followed by infertility in adults
  • What are the molecular mechanisms that produce reduced fertility after chemotherapy? "Risk of gonadotoxic chemoradiotherapy followed by infertility in adults" better explains.
  • Line 276: “The first part aimed to deter……………….” there is a word written with another character
  • Many anti-cancer drugs have adverse effects on the reproductive system. In particular, chemotherapy can alter the levels of hormones and sperm quality, leading to reduced fertility or infertility. Various chemotherapeutic agents have been categorised according to their gonadotoxicity into three risk categories: high-risk, medium-risk and low-risk chemotherapeutic agents. High-risk chemotherapeutic agents include the following alkylating agents: cyclophosphamide, busulphan, chlorambucil, procarbazine, melphalan, ifosfamide, chlormethamine. Medium-risk chemotherapeutic agents include platinum agents (cisplatin, carboplatin); anthracycline antibiotics (adriamycin [doxorubicin]); and taxoids (docetaxel and paclitaxel). Low-risk agents include vinca plant alkaloids (vincristine and vinblastine); anthracycline antibiotics (bleomycin) and antimetabolites (methotrexate, 5-fluorouracil, 6-MP [mercaptopurine]). Alkylating agents, the most gonadotoxic chemotherapeutic medications, cause dose-dependent, direct destruction of oocytes and follicular depletion, and may bring about cortical fibrosis and ovarian blood-vessel damage.

There are substances capable of playing a protective role against male reproductive toxicity

induced by heavy metals, environmental pollutants, and chemotherapy. This indicates that the molecular mechanisms in the induction of infertility may be similar. In this regard I suggest to read and quote the following works:

10.3390/ijms21186710

 10.3390/ijms21124198

  • Better explain all materials and methods

  • The English form in all the manuscript must be improved

Reviewer 2 Report

Journal: IJMS (International Journal of Molecular Sciences)

Manuscript ID: ijms-1562456

Manuscript Title: Immature testicular tissue engineered from weaned mice to adults for prepubertal fertility preservation- An in vivo translational study

Dear Authors,

The content of this manuscript is very important for basic research and clinical translation.

The methodology involved is sophisticated and highly original.

However, I would like that Authors could add some more information to facilitate reproducible results.

-Abstract

L32:

This surrogate model supported optimal spermatogenesis that can be achieved…

Introduction:

L40:

…are at risk of gonadotoxic chemoradiotherapy and are still experimental…

-Results:

L129:

The arrowhead shows the seminiferous tubules in fresh and scaffold groups…

Figure 4:

Labelling description is missing. Please indicate the type of germ cells achieved.

-Results:

Why Authors did not perform mating to observe offspring?

-Materials and Methods:

To reproduce results readers should have access to:

Number and diameter of testicular tissue allocated to the scaffold.

Which testicular tissue, seminiferous tubules?

Diameter of the scaffold transplanted. One or more scaffold. Free or linked. Grafted free in the scrotum or fixed to the skin.

Round 2

Reviewer 1 Report

The authors have responded to all my requests. I accept the manuscript in the present form